# Low Alanine Aminotransferase as a Marker for Sarcopenia and Frailty, Is Associated with Decreased Survival of Bladder Cancer Patients and Survivors—A Retrospective Data Analysis of 3075 Patients

**DOI:** 10.3390/cancers16010174

**Published:** 2023-12-29

**Authors:** Menachem Laufer, Maxim Perelman, Gad Segal, Michal Sarfaty, Edward Itelman

**Affiliations:** 1Department of Urology, Chaim Sheba Medical Center, Ramat Gan 5262112, Israel; 2Faculty of Medicine, Tel-Aviv University, Ramat Aviv, Tel Aviv 6139001, Israeledi.itelman@gmail.com (E.I.); 3Department of Internal Medicine “I”, Chaim Sheba Medical Center, Ramat Gan 5262112, Israel; 4Education Authority, Chaim Sheba Medical Center, Ramat Gan 5262112, Israel; 5Institute of Oncology, Chaim Sheba Medical Center, Ramat Gan 5262112, Israel; 6Department of Internal Medicine E, Rabin Medical Center, Beilenson Campus, Peta-Tiqva 4941492, Israel; 7Cardiology Division, Rabin Medical Center, Beilenson Campus, Peta-Tiqva 4941492, Israel

**Keywords:** bladder cancer, sarcopenia, frailty, alanine aminotransferase, survival

## Abstract

**Simple Summary:**

This article describes the results of a study that checked the association between low levels of ALT (an enzyme) in the blood and the outcomes of patients suffering from bladder cancer. Low ALT blood concentrations are known to be associated with a low, whole-body muscle mass. This state is defined as “sarcopenia” and is associated with frailty and worse outcomes. Indeed, we found that low ALT levels were associated with shortened survival. This finding is important for physicians when they plan the treatment for their patients. It is assumed that frail patients might experience worse results from aggressive management. This approach is called “personalized medicine”.

**Abstract:**

Background. Sarcopenia is characterized by the loss of muscle mass and function and is associated with frailty, a syndrome linked to an increased likelihood of falls, fractures, and physical disability. Both frailty and sarcopenia are recognized as markers for shortened survival in a number of medical conditions and in cancer patient populations. Low alanine aminotransferase (ALT) values, representing low muscle mass (sarcopenia), may be associated with increased frailty and subsequently shortened survival in cancer patients. In the current study, we aimed to assess the potential relationship between low ALT and shorter survival in bladder cancer patients and survivors. Patients and Methods. This was a retrospective analysis of bladder cancer patients and survivors, both in and outpatients. We defined patients with sarcopenia as those presenting with ALT < 17 IU/L. Results. A total of 5769 bladder cancer patients’ records were identified. After the exclusion of patients with no available ALT values or ALT levels above the upper normal limit, the final study cohort included 3075 patients (mean age 73.2 ± 12 years), of whom 80% were men and 1362 (53% had ALT ≤ 17 IU/L. The mean ALT value of patients within the low ALT group was 11.44 IU/L, while the mean value in the higher ALT level group was 24.32 IU/L (*p* < 0.001). Patients in the lower ALT group were older (74.7 vs. 71.4 years; *p* < 0.001), had lower BMI (25.8 vs. 27; *p* < 0.001), and their hemoglobin values were lower (11.7 vs. 12.6 g/dL; *p* < 0.001). In a univariate analysis, low ALT levels were associated with a 45% increase in mortality (95% CI 1.31–1.60, *p* < 0.001). In a multivariate model controlling for age, kidney function, and hemoglobin, low ALT levels were still associated with 22% increased mortality. Conclusions. Low ALT values, indicative of sarcopenia and frailty, are associated with decreased survival of bladder cancer patients and survivors and could potentially be applied for optimizing individual treatment decisions.

## 1. Background

### 1.1. Personalized Medicine for Cancer Patients and Survivors

Clinical tools for the assessment of a patient’s physiological status are important for treatment decision-making and estimation of prognosis for both cancer patients and cancer survivors. This paradigm is called “personalized medicine”. The American Cancer Society and others do not clearly distinguish between precise and personalized medicine, stating both as “a way health care providers can offer and plan specific care for their patients, based on the particular genes, proteins, and other substances in a person’s body” [1,2] others make a difference: while precision medicine focuses on pathologic elements and disease molecular and histological characteristics, personalized medicine focuses on aspects of the human physiology and the pre-morbid background of patients [3,4]. Means and practical clinical tools for such a “gestalt” approach to patients are advocated and used: the Eastern Cooperative Oncology Group performance status (ECOG-PS) and Charlson comorbidity index (CCI) in surgical candidates are commonly used tools [5,6]. However, those tools are generally based on caregivers’ subjective assessments and are more difficult to define and standardize. Amongst other tools for treatment personalization, there are several tangible, quantitative measures for personalized patient appreciation that target the realm of sarcopenia and frailty.

### 1.2. Sarcopenia and Frailty in the Realm of Personalized Oncology

Sarcopenia is characterized by the loss of muscle mass and function and is associated with frailty, a syndrome that leads to an increased likelihood of falls, fractures, physical disability, recurrent hospitalizations, and mortality [7,8,9,10]. In cancer patients, recognition of sarcopenia and frailty improves the selection of robust patients eligible for more aggressive therapies such as major surgery and chemotherapy [11,12]. Both frailty status and sarcopenia have been identified as markers for shortened survival in cancer patients [13]. The assessment of sarcopenia and frailty could, therefore, be a way towards personalized medicine.

### 1.3. The Case for Bladder Cancer Patients

In the year 2020, bladder cancer was diagnosed in approximately 573,000 patients worldwide, with 213,000 reported deaths. It is four times more common in men than in women, making it the 6th most common cancer in men. Smoking is a major risk factor for bladder cancer, and other factors, such as occupational exposure, may be major causes in some populations [14]. Bladder cancer more commonly affects elderly patients with significant comorbidity and lower performance status [15], making these patients highly prone to the devastating effects of sarcopenia and frailty. Detection and follow-up in bladder cancer are currently based on cystoscopy, but data on markers are promising [16]. Non-invasive tumors are treated with endoscopic resection and intravesical therapy, whereas in patients with muscle-invasive disease, radical cystectomy (RC) is the gold standard of treatment [17]. Platinum-based chemotherapy is often used in neoadjuvant, adjuvant, and metastatic settings [18]. Subjective measures such as comorbidity and performance status predict surgical outcomes as well as shortened survival after RC [19,20]. Several studies investigated more objective parameters, including sarcopenia and frailty in bladder cancer patients. Both are prevalent in bladder cancer and are usually correlated with adverse surgical outcomes and shortened survival [21,22,23,24,25,26,27,28,29,30]. There is still a need for an available biomarker that could differentiate between frail and robust bladder cancer patients. This tool should be readily available and enable good, independent differentiation between these groups of bladder cancer patients.

### 1.4. Alanine Aminotransferase as a Biomarker for Sarcopenia and Frailty

Alanine aminotransferase (ALT) is an intracellular enzyme, abundant in the liver parenchyma, commonly monitored in routine bloodwork. Its biochemical activity catalyzes pyruvate to alanine in the skeletal muscle and alanine to pyruvate in the liver [31]. As such, ALT could transform potential energy within muscle cells whenever pyruvate is not fully assimilated into the Krebs cycle by exporting alanine to the lever, where it is once again transformed by ALT to pyruvate and exploited as available energy. Elevated ALT blood levels are used as a diagnostic marker of liver-tissue damage (hepatitis), but little was known until recently regarding the clinical consequences of low -normal ALT blood levels. During the past several years, our study group has repeatedly demonstrated that a low ALT activity in the peripheral blood (measured in international units per liter), representing low muscle mass (sarcopenia), is associated with shortened survival in middle-aged adults [32] and in patients hospitalized for various causes [33,34]. In addition, low ALT values were found to be associated with increased frailty and shortened survival in various cancer patient populations [35]. The association between low ALT values, sarcopenia and frailty parameters, and shortened survival were persistently found to be statistically significant and independent of age, nutritional status, kidney functions, and significant comorbidities. Therefore, low-normal ALT levels can be used as a surrogate marker for sarcopenia and frailty and could be used as simple, accessible tools for clinicians in the mission of prognosis and treatment plans’ personalization.

In the current study, we assessed the possible association between low serum ALT, indicative of sarcopenia and frailty, and shorter survival in a large cohort of bladder cancer patients and survivors. Our aim was to offer a new tool for urologists and oncologists treating these patients to be assimilated into their clinical routines as a tool of personalized medicine.

## 2. Patients and Methods

### 2.1. Patients’ Characteristics

The current study included men and women diagnosed with bladder cancer who were treated in a large, tertiary medical center as outpatients or inpatients. Patients underwent either surgery, chemotherapy, immunotherapy, or local treatment. Following approval by the local ethics committee (IRB approval # SMC-23-0737), including waiver on informed consent as the study was retrospective, all patients’ characteristics, demographics, and other relevant clinical data were retrieved from their electronic medical records. These are the same medical records used for routine clinical management and are, therefore, considered a reliable source of clinical information. We excluded patients with ALT levels higher than 40 IU (our institutional upper limit of normal ALT values) that are primarily associated with damaged liver tissue (various types of acute and chronic hepatitis); therefore, they could no longer be considered as a reliable marker for striated muscles’ mass. The final cohort includes patients with ALT levels that were established at the time of bladder cancer diagnosis. The primary outcome of the current study was all-cause mortality. Survival data were available for all subjects from the Israeli Population National Registry, which is updated four times annually.

### 2.2. Descriptive Statistics and Analytical Methods

Continuous variables, when normally distributed, are shown as mean ± standard deviation (SD) or median with interquartile range (IQR) whenever skewed. We determined the normality of variables using the Anderson-Darling and Shapiro–Wilk tests. Categorical variables were presented as frequency (%). We compared continuous data with a student’s *t*-test, and categorical data were compared using chi-square or Fisher exact tests. Survival was analyzed using a Log-rank test and depicted using a Kaplan–Meier curve. We used univariate Cox regression modeling to determine the unadjusted Hazard Ratio (HR) for the primary outcome and a multivariate model for the correlation and control for possible confounders. An association was considered statistically significant for a two-sided *p* value lower than 0.05. Analyses were performed using R software version 4.1.0 (R Foundation for Statistical Computing).

## 3. Results

A total of 5769 bladder cancer patients’ records were identified, of whom 2694 patients were excluded due to missing ALT values at baseline or ALT levels above the upper normal limit of 40 IU/L. The final study population included 3075 patients, of whom 1632 would be classified as patients with sarcopenia due to baseline ALT levels lower than 17 IU/L. the cutoff point at 17 IU/L was predetermined according to results and reports by previous clinical studies. Figure 1 details our patients’ consort flow and exclusion diagram of this study.

The mean age for the entire cohort was 73.2 ± 12 years, and 80% were men. Mean ALT levels for the whole study population were 17.49 IU/L, and 1362 patients (53%) had low ALT levels, defined as levels lower than 17 IU/L. Patients’ demographics and characteristics are detailed according to their ALT levels (lower or ≥ 17 IU/L) in Table 1: patients with lower ALT values were older (74.7 ± 11.8 vs. 71.4 ± 12.1 years; *p* < 0.001) and had lower body mass index (BMI) values (25.8 ± 4.6 vs. 27 ± 4.6; *p* < 0.001). Both age and BMI correlations with lower ALT values are logical and reproduced in previous studies and, therefore, serve as a reassurance for the authenticity of data. While there were no significant differences in their background diagnoses (e.g., arterial hypertension, diabetes mellitus, chronic obstructive heart disease, and stroke), patients with lower ALT values had lower hemoglobin levels (11.76 ± 2.15 vs. 12.61 ± 2.12 g/dL; *p* < 0.001) and higher baseline creatinine levels (1.44 ± 1.17 vs. 1.28 ± 0.9 mg/dL; *p* < 0.001)—both, also, are recurring evidence in previous studies.

### 3.1. Univariate Analysis

In a univariate analysis, ALT levels lower than 17 IU/L were associated with a 45% increase in mortality (HR = 1.45, 95% CI 1.31–1.60, *p* < 0.001). Figure 2 shows a Kaplan-Meir curve for the crude survival analysis according to ALT levels. The separation of curves occurred early along the follow-up and was continuously statistically significant.

### 3.2. Multivariate Analysis

In a multivariate model (Table 2) controlled for age, creatinine levels, and albumin, low ALT levels were still associated with 22% increased mortality in a statistically significant manner (HR = 1.22, 95% CI 1.10–1.35, *p* < 0.001). Both age (per each year), serum creatinine, albumin blood concentration, presence of arterial hypertension, and diabetes mellitus were demonstrated as independent risk factors of shortened survival.

## 4. Discussion

Clinical tools that are used for assessing a patient’s physiological reserves are crucial for clinical decision-making in both medical and surgical realms. They should be employed to better assess the prognosis and treatment planning for cancer patients and survivors. In the most updated version of the NCCN (National Cancer Comprehensive Network) guidelines for bladder cancer patients, it is stated that: “No single follow-up plan is appropriate for all patients. Follow-up frequency and duration should be individualized based on patient requirements, and may be extended beyond 5 years after shared decision-making between the patient and physician.” Yet, the terms personalized medicine or patients’ robustness are not included in the document [36]. The terms “sarcopenia” and “frailty” are also not mentioned in the ESMO, European Society of Medical Oncology guidelines for genitourinary malignancies [37], although the aforementioned should be considered as basics of individualized patient care.

Both frailty and sarcopenia have been identified as markers for shortened survival in patients with various types of cancers. Sarcopenia is a condition characterized by the loss of muscle mass and function and is often one of the most important hallmarks of cancer cachexia [38], with over one hundred studies that evaluated the association between lean muscle mass and cancer mortality. The overall pooled HR on cancer mortality was 1.69 (95% CI, 1.56 to 1.83) for patients with sarcopenia [3]. Today, “eyeballing” the patient serves most clinicians in the appreciation of patients’ robustness. Multiple scientific/objective tools and methods have been offered to detect and classify sarcopenia, but none showed a clear superiority. Body mass index (BMI), another common tool used for general health assessment, and body surface area (BSA) are both widely used but do not reflect lean muscle mass [39]. In reference to other components of the metabolic syndrome, Crocetto et al. describe the ratio of triglycerides to High-Density Lipoprotein (HDL) and the extent of Pseudocholinesterase (PChE) activity as potential markers for Bladder cancer existence and their potential roles as screening for the disease [40]. Measurement of L3SMI (the striated muscle mass at the level of the third lumbar vertebra (by computed tomography) has been a recommended method to detect sarcopenia [41,42,43], with the L3 psoas index has been proposed as a simplified alternative [44]. Nevertheless, this approach has yet to be further validated. In their review [45], Williams et al. differentiated sarcopenia in cancer patients from cancer-related cachexia by the following parameters: while decreased muscle mass is the whole mark of sarcopenia, it is not necessarily so in the case of cachexia. Cachexia is defined as a reduction in the overall body weight (while sarcopenia is not) and a decline in the body fat mass. While in sarcopenia, there is a decrease in the basal metabolic rate, it is increased in patients suffering from cachexia. Lastly, while cachexia is characterized by increased markers of inflammation, patients with sarcopenia will not present as such.

In parallel to sarcopenia, frailty is more elusive. Frailty is defined as a state of vulnerability for poor return to homeostasis following acute illness, surgery, and chemotherapy [46]. Frail cancer patients are known to be at increased risk of postoperative complications, chemotherapy side effects, other morbidity, and death. Over 70 different frailty measures are poorly validated, and they range from a single-item assessment, such as gait speed or sarcopenia, to a comprehensive questionnaire with more than 90 items [47]. Frailty should be considered as the phenotypic/functional translation of sarcopenia. Williams et al. state that frail patients, in contrast to those with mere sarcopenia, will show decreased overall body weight, while decreased basal metabolic rate is a common feature of both sarcopenia and frailty [45].

A possible surrogate marker for sarcopenia and frailty is low serum activity of ALT. ALT values that are within the normal range (5 to 40 IU/L) represent the striated muscle mass in the body rather than liver cells’ damage, as it is appreciated whenever ALT levels are above the upper normal limits. Several studies have demonstrated that low ALT activity is associated with shortened survival in various patient populations [32,33,34]. Low ALT values, representing low muscle mass (sarcopenia), may be associated with increased frailty and subsequently shortened survival in various cancers [35,48,49].

Sarcopenia is common in bladder cancer patients. The majority of patients with bladder cancer are in their 7th decade or older and, therefore, are at greater risk for sarcopenia and frailty. Additionally, many patients undergo radical cystectomy, major abdominal surgery, and urinary diversion that is associated with profound metabolic consequences, including sarcopenia [23,24,25]. Systemic chemotherapy and immunotherapy (immune checkpoint inhibitors) are very commonly used in bladder cancer patients either in the neoadjuvant setting (before cystectomy) or in advanced stages. Those treatments might also contribute to the accelerated appearance of sarcopenia and frailty [12,50].

Based on the findings presented in this manuscript, we aim to include sarcopenia and frailty in the decision-making processes regarding bladder cancer patients, using low ALT values as a marker for sarcopenia, frailty, and shortened survival. It may be applicable to patients who are candidates for radical surgery, medical treatments for advanced and metastatic disease, and possibly also for older patients with early-stage disease. Others have published on this matter: Using standard sarcopenia metrics, Fukushima et al. [24], in a systematic literature review, found that sarcopenia was associated with shorter survival both in radical cystectomy patients and very advanced inoperable disease. Ibilibor et al. [18] reported similar results in a comprehensive review of patients undergoing radical cystectomy. However, there is significant heterogeneity across the studies in terms of the definition of sarcopenia. Frailty is another well-established prognostic factor in bladder cancer patients who are candidates for radical surgery [27]. Our study confirms these results but is unique in 3 ways: First, it involves more than 3000 patients. Secondly, the results apply to the whole spectrum of bladder cancer patients and not just to radical cystectomy or very advanced cancer patients. Lastly, it is based on a routine, simple blood test: ALT activity values that are part of every routine chemistry analysis and are almost always available for the attending urologists and oncologists.

Several authors reported on serum AST to ALT (De Ritis) ratio as a predictor of survival in bladder cancer, mostly in radical cystectomy patients [51,52,53]. However, the theory behind this ratio in bladder cancer is that it represents hepatic glucose metabolism and its relation to high glucose turnover in bladder cancer patients. Low ALT values, as referred to in our study, are a possible surrogate for sarcopenia and frailty and seem to better present a different and well-established pathway.

Mori et al. [54] published an extensive review on other blood-based, possible predictors of mortality after radical cystectomy, with thirty-two studies including 22,224 patients that were found eligible for inclusion into the meta-analysis: Several preoperative biomarkers were significantly associated with cancer-specific survival: Three are inflammatory markers: increased neutrophil to lymphocyte ratio (HR: 1.20, 95% CI 1.11–1.29), increased levels of C-reactive protein (HR: 1.44, 95% CI 1.26–1.66) and the increase in white blood cell (WBC) count (HR: 1.05, 95% CI 1.02–1.07). Two more markers in this review were found to be related to the temporal, nutritional status of patients: low hemoglobin concentrations (HR: 0.87, 95% CI 0.82–0.94) and low albumin to globulin ratio (HR: 0.26, 95% CI 0.14–0.48). Compared with other blood-based biomarkers, low ALT in our study (lower than 17 IU/L, a value established in several previous studies) seems to be a powerful marker for survival, with a significant 45% increase in mortality. Low ALT is a surrogate for sarcopenia and frailty, which are well-documented predictors of prognosis in bladder cancer. In addition, Low ALT-associated shortened survival applies to all bladder cancer patients and survivors unrelated to radical cystectomy. In contrast to the above-mentioned biomarkers, we relate to low ALT levels as a reliable representative of sarcopenia and frailty, while hemoglobin and albumin levels should be related to as biomarkers for the temporal nutritional status of patients rather than reflecting their status of sarcopenia and frailty.

## 5. Study Limitation

This was a single-center, retrospective study. Future prospective studies are suggested in order to establish the role of sarcopenia and frailty estimation amongst bladder cancer patients by measurement of their baseline ALT levels.

We examined ALT activity in patients who received systemic chemotherapy and/or immunotherapy for advanced or metastatic cancer in comparison to the full cohort. Low ALT was associated with decreased survival in both cohorts. Staging data was not fully available and, therefore, was not included. Bladder cancer subtypes are rare in our cohort; 95% are urothelial cancer. As stated in the conclusion, the role of ALT should be in the realm of personalized medicine rather than in a complementary manner to the realm of precision medicine.

## 6. Conclusions

Advanced, precision-medicine tools such as staging, genetic sub-typing, and prospective results of chemotherapy and immunotherapy, already used by clinical oncologists, should be accompanied by better tools for personalization of treatment options, especially in elderly cancer populations. Sarcopenia and frailty, representing the gestalt of patients’ robustness and ability to withstand both severe disease and aggressive therapies—should be incorporated into clinical decisions. The current study demonstrated that low serum ALT activity (lower than 17 IU/L) is indicative of sarcopenia and frailty and is associated with reduced survival of bladder cancer patients and survivors. Therefore, baseline ALT measurements, which are commonly taken for all patients, should be incorporated into the treatment decisions in this patient population.

## Figures and Tables

**Figure 1 cancers-16-00174-f001:**
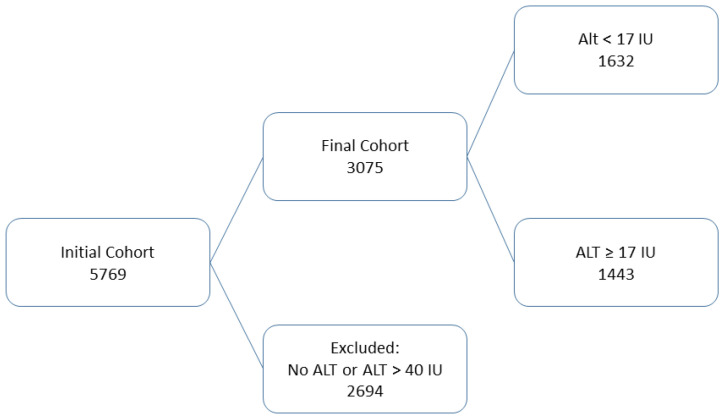
Consort flow diagram of patients.

**Figure 2 cancers-16-00174-f002:**
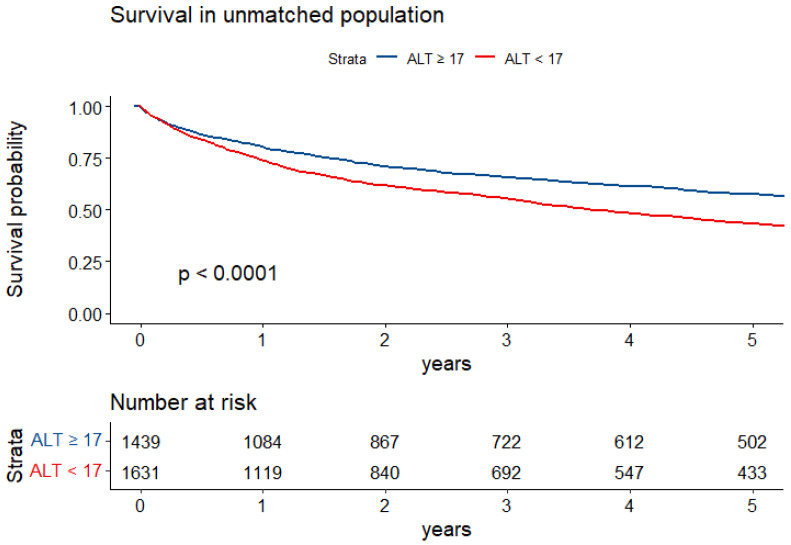
Kaplan Meir Survival Analysis according to ALT levels.

**Table 1 cancers-16-00174-t001:** Patients’ characteristics according to ALT level.

	Overall, N = 3075	ALT ≥ 17, N = 1443	ALT < 17, N = 1632	*p* Value
ALT baseline; Mean (SD)	17.49 (8.06)	24.32 (6.29)	11.44 (3.06)	<0.001
Patients’ Demographics				
Age [years; Mean (SD)]	73.2 (12)	71.4 (12.1)	74.7 (11.8)	<0.001
Body-mass index (BMI) [Mean (SD)]	26.4 (4.6)	27 (4.6)	25.8 (4.6)	<0.001
Male Gender N (%)	2474 (80)	1195 (83)	1279 (78)	0.002
Background diagnoses				
Arterial hypertension—N (%)	1778 (58)	822 (57)	956 (59)	0.386
Diabetes Mellitus—N (%)	958 (31)	440 (30)	518 (32)	0.48
Dyslipidemia—N (%)	1280 (42)	628 (44)	652 (40)	0.049
Ischemic Heart Disease—N (%)	971 (32)	458 (32)	513 (31)	0.886
Chronic Obstructive Pulmonary Disease—N (%)	365 (12)	176 (12)	189 (12)	0.638
Atrial Fibrillation-Flutter—N (%)	384 (12)	171 (12)	213 (13)	0.342
Stroke—N (%)	309 (10)	145 (10)	164 (10)	1
Laboratory parameters				
Albumin g/dL—Mean (SD)	3.81 (2.42)	3.83 (1.59)	3.8 (2.97)	0.686
Hemoglobin g/dL—Mean (SD)	12.16 (2.18)	12.61 (2.12)	11.76 (2.15)	<0.001
Creatinine mg/dL—Mean (SD)	1.37 (1.05)	1.28 (0.9)	1.44 (1.17)	<0.001

**Table 2 cancers-16-00174-t002:** Multivariate Analysis.

	HR (95% CI)	*p* Value
ALT < 17 IU/L	1.22 [1.10, 1.35]	<0.001
Age	1.04 [1.04, 1.05]	<0.001
Serum Creatinine	1.20 [1.16, 1.25]	<0.001
Albumin	0.78 [0.71, 0.86]	<0.001
Hypertension	0.89 [0.80, 0.99]	0.029
Diabetes Mellitus	1.15 [1.03, 1.28]	0.011

## Data Availability

All data supporting the findings of this study are kept by the principal investigator. Data will become available by appropriate requests.

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
