# Peer review of "Low Alanine Aminotransferase as a Marker for Sarcopenia and Frailty, Is Associated with Decreased Survival of Bladder Cancer Patients and Survivors—A Retrospective Data Analysis of 3075 Patients"

_cancers, 2023, doi:10.3390/cancers16010174_

Round 1

Reviewer 1 Report

Comments and Suggestions for Authors

It is my pleasure to review this paper entitled “Low Alanine Aminotransferase as a Marker for Sarcopenia and Frailty, is Associated with Decreased Survival of Bladder Cancer Patients and Survivors. A Retrospective Data Analysis of 3075 Patients”. The aim of this paper is to evaluate the possible association between low serum ALT, and shorter survival bladder cancer patients and to offer a new tool for urologists and oncologists, for treating these patients. However, there are some drawbacks that could be addressed before an eventual publication.

Introduction should be revised it is too much long.

Authors should highlight the strengths and limitations of the paper.

Line 200 please clarify the meaning of L3SMI.

Author Response

Attached please find a point-by-point answers to review.

On behalf off all authors I thank you very much.

Gad Segal, MD. 

Reviewer 2 Report

Comments and Suggestions for Authors

I would congratulate the authors for their work and for addressing an important topic. Only some points warrant mention:

1.    In the “Introduction” section, I suggest to improving this section regarding recent research on serum markers useful in the early detection and diagnosis of bladder cancer

2.    In the “Results” section, “Figure 2” should be improved by adding the numerosity of the population at risk, otherwise it cannot be defined as a Kaplan-Meier curve.

3.    In the “Discussion” section, when discussing on alteration of BMI, the authors should also include a brief mention of the opposite case, such as obesity, and its associated alteration of triglycerides and cholesterol and BCa, such as in PMID: 35204522.

4.    In the “Discussion” section, the term “and co.” should be replaced with “et al.”

Author Response

(The authors gave the same response as above.)

Round 2

Reviewer 1 Report

Comments and Suggestions for Authors

I want to thanks the authors for the changes

Author Response

We thank you very much